# Four Cholesterol-Recognition Motifs in the Pore-Forming and Translocation Domains of Adenylate Cyclase Toxin Are Essential for Invasion of Eukaryotic Cells and Lysis of Erythrocytes

**DOI:** 10.3390/ijms23158703

**Published:** 2022-08-05

**Authors:** Jone Amuategi, Rocío Alonso, Helena Ostolaza

**Affiliations:** 1Department of Biochemistry and Molecular Biology, Faculty of Science and Technology, University of the Basque Country UPV/EHU, 48080 Bilbao, Spain; 2Instituto Biofisika (UPV/EHU, CSIC), University of the Basque Country, 48940 Leioa, Spain; 3Fundación Biofísica Bizkaia (FBB), Barrio Sarriena s/n, 48940 Leioa, Spain

**Keywords:** bacterial toxins, RTX toxins, pore-forming toxins, lipid-protein interactions

## Abstract

Adenylate Cyclase Toxin (ACT or CyaA) is one of the important virulence factors secreted by *Bordetella pertussis*, the bacterium causative of whooping cough. ACT debilitates host defenses by production of unregulated levels of cAMP into the cell cytosol upon delivery of its N-terminal domain with adenylate cyclase activity (AC domain) and by forming pores in the plasma membrane of macrophages. Binding of soluble toxin monomers to the plasma membrane of target cells and conversion into membrane-integrated proteins are the first and last step for these toxin activities; however, the molecular determinants in the protein or the target membrane that govern this conversion to an active toxin form are fully unknown. It was previously reported that cytotoxic and cytolytic activities of ACT depend on membrane cholesterol. Here we show that ACT specifically interacts with membrane cholesterol, and find in two membrane-interacting ACT domains, four cholesterol-binding motifs that are essential for AC domain translocation and lytic activities. We hypothesize that direct ACT interaction with membrane cholesterol through those four cholesterol-binding motifs drives insertion and stabilizes the transmembrane topology of several helical elements that ultimately build the ACT structure for AC delivery and pore-formation, thereby explaining the cholesterol-dependence of the ACT activities. The requirement for lipid-mediated stabilization of transmembrane helices appears to be a unifying mechanism to modulate toxicity in pore-forming toxins.

## 1. Introduction

Pore Forming Toxins (PFTs) constitute a very particular group of proteins that are secreted as soluble monomers and target the plasma membrane of eukaryotic cells where become membrane-embedded proteins to exert their lytic effect [1,2,3,4]. This means that hydrophobic regions of the toxin that are initially protected from the aqueous medium must be exposed once in contact with the membrane to allow their insertion into the lipid bilayer and subsequent pore formation [1,2,3,4]. This transition from a water-soluble structure to a distinct membrane-associated protomer involves structural rearrangements of the protein in the membrane environment. Membrane lipids are important regulators that can influence the process of proteins insertion in two ways, indirectly, by modulating the biophysical properties of the lipid bilayer such as fluidity, phase segregation, bilayer thickness, tension, etc., which may affect protein structure and function [5,6,7], or directly, through specific lipid-protein interactions [8,9,10].

Adenylate cyclase toxin (ACT, CyaA, or AC-Hly) is a 1706 residue-long pore-forming leukotoxin secreted by pathogenic *Bordetellae* and plays a key role in virulence [11,12,13]. ACT belongs to the family of Repeats-in-ToXin (RTX) protein toxins exhibiting cytotoxic/cytolytic pore-forming activity [14,15] and consists of an N-terminal enzymatic adenylate cyclase (AC) domain of ≈400 residues that is fused by an “AC to Hly linker segment” of about 100 residues so-called translocation region (TR), to a pore-forming RTX hemolysin (Hly) moiety of approximately 1200 residues. The Hly hemolysin moiety itself consists of a hydrophobic pore-forming domain (HD), a fatty acyl-modified domain in which two conserved Lys residues are covalently acylated, an RTX calcium-binding domain and a C-terminal secretion signal [14,15].

Cytotoxicity by ACT on target cells results from the generation by its AC domain of unregulated levels of cAMP in the target cell cytosol and the permeabilization of the plasma membrane by oligomeric pores formed by its RTX hemolysin domain [15]. Hly mediates cell binding of ACT to the target membrane and then delivers the enzymatic AC domain into the cytosol of host cells, where the AC binds calmodulin and catalyzes the conversion of intracellular ATP to cAMP, thereby subverting cellular signalling [16,17]. Besides elevating cAMP, ACT exhibits a modest haemolytic activity due to its capacity to form cation-selective pores that permeabilize cellular membrane and eventually provoke osmotic cell lysis [18,19,20].

Numerous studies have highlighted the importance of various amino acid residues and/or the contribution of distinct domains of the ACT polypeptide for the translocation of the AC domain and for the lytic activity [21,22,23,24,25,26,27]. However, a description at molecular level of the individual steps followed by the ≈40 kDa AC domain polypeptide to cross the membrane or by the Hly domain to form oligomeric lytic pores is still missing.

Two ACT regions, the translocation region (TR) located between amino acids ≈400 to 500 and the hydrophobic domain (HD) that extends from residues ≈500 to 700, have been directly implicated in AC domain translocation and pore formation [21,22,23,24,25,26,27,28]. Upon membrane binding, several amphipathic/hydrophobic α-helices (HI to HV) of the hydrophobic domain insert into the plasma membrane building the pore structure [21,22,23,24,25,26,27]. AC domain translocation requires, apart from the HD insertion, the interaction with the membrane of other two long α-helices outside the pore-forming domain, predicted to form between residues ≈413 to 434 and ≈454 to 484 (translocation region) [28,29]. So far, it is unknown whether all these helical elements insert spontaneously into the membrane or whether their embedding into the lipid bilayer may require assistance from membrane lipids.

At that regard, it was reported in previous studies that AC translocation depends on membrane cholesterol content, in the sense that a decrease of cholesterol yielded a significant decrease in the capacity of ACT to translocate the AC domain across cell membrane [30]. More recently, we found that presence of cholesterol in the membrane enhances the lytic capacity of ACT on erythrocytes and artificial membranes by promoting toxin insertion and oligomerization [31]. Similar results were reported for the HlyA toxin, showing that incorporation of cholesterol into phospholipid bilayers promoted the irreversible insertion of the toxin into the membrane, which increased the toxin lytic activity [32]. Moreover, three other pore-forming toxins of the same RTX family, the *Aggregatibacter actinomycetemcomitans* leukotoxin LtxA, the *Escherichia coli* hemolysin HlyA, and the *Kingella kingae* RtxA cytolysin were previously shown to specifically bind cholesterol [33,34,35], suggesting that requirement of cholesterol might be other common identity sign shared by the RTX toxins.

Here we decided to investigate whether the cholesterol dependence of the lytic and translocation activities of ACT is due to direct, specific binding of ACT to membrane cholesterol, and to explore the possible molecular determinants. With this aim we took a closer look at two types of cholesterol binding motifs, the so-called cholesterol recognition/interaction amino acid consensus (CRAC) motif with the **L/V**-X(1–5)-**Y/F**-X(1–5)-**R/K** pattern, and the reverse CARC motif with the **R/K-**X(1–5)-**Y/F**-X(1–5)-**L/V** pattern [36,37,38]. CRAC and CARC sites have been identified in transmembrane segments of several membrane proteins, such as G-protein coupled receptors, but also in membrane-interacting segments of several bacterial toxins and viral proteins that interact with cholesterol [39,40,41,42]. Moreover, in the aforementioned three RTX toxins (LtxA, HlyA and RtxA), CRAC sites have been identified in their pore-forming domains and for two of them, LtxA and RtxA, the interaction with cholesterol has been experimentally demonstrated [33,34,35]. Single residue substitutions in the central aromatic residue (Tyr or Phe) of CRAC and CARC motifs have been found to strikingly reduce or eliminate protein-cholesterol interactions in different cholesterol-binding proteins, affecting consequently protein activity in membranes.

We have constructed several site-directed mutants to examine the role of four potential CRAC and CARC motifs (CARC^415^, CRAC^485^, CRAC^521^ and CARC^532^) that could be predicted in the sequence of the membrane-interacting translocation region and hydrophobic domain of ACT, as possible molecular determinants of ACT-cholesterol interaction. By using site-directed mutants and functional studies, we show that the four motifs are functional cholesterol-binding sites and essential for the AC domain translocation and for the lytic activity on erythrocytes. We hypothesize that binding to cholesterol through these sites drives the insertion and proper transmembrane topology of α-helices in the ACT pore forming and translocation domains necessary for ACT activity.

## 2. Results

### 2.1. Specificity in the Interaction of ACT with Cholesterol

To corroborate that cholesterol dependence of the lytic and translocation activities of ACT is due to direct binding of ACT to membrane cholesterol, and not, to indirect effects of the sterol on the physical state of the phospholipid bilayer, we performed different experiments. First, we quantified ACT binding to liposomes of different lipid compositions: pure dioleylphosphatidylcholine (DOPC), DOPC:Chol (3:1 molar ratio) and DOPC:Erg (3:1 molar ratio).

Data depicted in Figure 1 show that binding of ACT to pure lipid vesicles was notably greater in presence of cholesterol relative to the pure phospholipid liposomes, and that ergosterol (ergosta-5,7,22-trien-3β-ol), an analogue of cholesterol, could not reproduce this effect.

In other set of experiments, the toxin (100 nM) was preincubated with increasing concentrations of free cholesterol for 30 min at RT and was then further incubated with erythrocytes in the presence of free cholesterol, and haemolysis was measured. As shown in Figure 2, free cholesterol at concentrations above 5 µM had a notable inhibitory effect on the toxin-induced erythrocyte lysis, suggesting that ACT may recognize and directly bind to membrane cholesterol.

This idea was reinforced by other experiment in which ACT (50 nM) was first pre-incubated with liposomes of different lipid composition (DOPC, DOPC:Chol (3:1 molar ratio) and DOPC:Erg (3:1 molar ratio), at two different final lipid concentrations (0.1 and 1 mM), and then assayed for haemolysis. The rationale of the experiment is that the more toxin is bound first to the liposomes, the less will remain unbound to bind then to the erythrocytes, and so the lower will be the haemolysis observed. Data in Figure 3 illustrate this process, showing that the cholesterol-containing liposomes are the most effective in binding ACT molecules, since it is this incubation, which yields the lower percentage of haemolysis. It is also shown that the reduction in haemolysis induced by the presence of ergosterol in the liposomes is very similar to the induced by the DOPC vesicles, consistent with the similar, lower ACT binding to these two cholesterol-free liposomes (Figure 1).

DOPC/Chol mixtures are completely miscible in all proportions up to the solubility limit of Chol, which is approximately 66%. Above that concentration, Chol precipitates as monohydrate crystals coexisting with DOPC/Chol membrane [43]. Therefore, the mixtures used in these assays, which have 25% cholesterol (DOPC/Chol 3:1, molar ratio), have a single phase, type Ld. Excluded therefore, a possible effect of cholesterol on the physical state of the lipids as effector of the increased ACT binding, we confirmed thus a specific ACT-cholesterol interaction.

### 2.2. Potential Cholesterol-Recognition Motifs Can Be Identified in the Sequence of the Membrane-Interacting Translocation Region and Hydrophobic Domain of ACT

Five helical elements (HI_502–522_, HII_527–550_, HIII_571–592,_ HIV_607–627_ and HV_678–698_) in the hydrophobic domain extending from residues ≈500–700 are believed to insert into the cell membrane, building the ACT pore structure [21,22,23,24,25,26,27]. Other two helices out of the pore-forming domain, h1 (residues 418–439) and h2 (residues 454–484), that constitute the translocation region, interact with the lipid bilayer for AC domain translocation [28,29]. Thus, we performed our search for potential cholesterol-binding CRAC and CARC motifs in the sequence of these membrane-interacting ACT domains. For that the EMBROSS: fuzzpro software program (http://emboss.bioinformatics.nl/cgi-bin/emboss/fuzzpro, accessed on 24 February 2020) was run. Sequences given as a search pattern were: [LV]-X(1,5)-**Y/F**-X(1,5)-[RK], [RK]-X(1,5)-**Y/F**-X(1,5)-[LV].

The analysis revealed eight putative cholesterol-binding motifs (Table 1). Four of the motifs identified were CRAC motifs with central Tyr tyrosine residues (Y632, Y658, Y725 and Y738) located at the end of the pore-forming domain, near the HIV and HV helices. In a previous study, other group reported that CRAC-disrupting substitutions of those four tyrosine residues had no impact on toxin activities, concluding that the four Tyr-containing CRAC motifs are not involved in cholesterol binding [44]. Thus, we focused our study on the other four motifs identified to check whether they were the molecular determinants of cholesterol interaction by ACT.

Two of the motifs are at the translocation region: CARC^415^ motif (residues 413–420, RS**F**^415^SLGEV) is located at the N-terminus of the long α-helix h1, and CRAC^485^ motif (residues 481–487, LMTQ**F**^485^GR) is located at the C-terminus of the second α-helix, h2. The other two motifs are localized at the pore-forming domain, CRAC^521^ motif (residues 518–527, VSG**F**^521^FR) localizes at the C-terminus of the first α-helix, HI, and CARC^532^ motif (residues 527–534, RWAGG**F**^532^GV) is located at the N-terminus of the second α-helix, HII (see Figure 4).

### 2.3. Point Mutation of the Central Phe Residue in the CRAC and CARC Motifs Have a Differentiated Effect on the ACT-Induced Haemolysis

To determine whether these four CRAC/CARC motifs could be molecular determinants of the ACT-cholesterol interaction and thereby of the cholesterol dependence of the lytic and translocation activities of ACT, we selectively mutated key aromatic amino acid residues in these motifs. Mutations in the central Tyr or Phe residues in CRAC and CARC motifs have been shown to strikingly reduce or eliminate protein-cholesterol interactions in different cholesterol-binding proteins, affecting consequently protein activity in membranes. We constructed four single mutant proteins with alanine, Ala substitutions in the central phenylalanine, Phe residues 415, 485, 521 and 532 of the respective motifs. Then we checked firstly the effect of these mutations on the toxin-induced haemolysis.

Figure 5 shows the raw traces of the kinetics recorded from a representative experiment of haemolysis induced by wild type ACT (50 nM) or by each one of the mutant toxins (50 nM), namely F415A, F485A, F521A and F532A mutants.

From haemolytic kinetics such as the observed in Figure 5, data at 180 min were represented in Figure 6A. In addition, the t_1/2_ values (time required to induce 50% haemolysis) were plotted in Figure 6B. As observed in the figures, the effect of the mutations was different depending on the location of the CRAC/CARC sites in the ACT structure. The individual substitutions of Phe by Ala, in the respective CRAC^521^ and CARC^532^ motifs (F521A and F532A) at the HD, induced a prominent inhibitory effect in the lytic activity of the mutant toxins, in both cases slowing down the erythrocytes lysis (Figure 6A) and reducing to the half the maximum haemolysis extent after 180 min of incubation (Figure 6B). In contrast, the Phe→Ala substitutions in the CARC^415^ and CRAC^485^ motifs (F415A and F485A) led to a faster and a greater lytic activity of the respective mutant toxins. This was reflected in the significantly greater maximum haemolysis values obtained after 180 min incubation (Figure 5 and Figure 6A), and in the lower t_1/2_ values (time in minutes required to induce 50% haemolysis) (Figure 6B).

To determine whether such mutations had any effect on toxin binding to lipid bilayers we performed a control experiment. Data represented in Figure 7 indicated that the binding percentage was similar for the four mutant toxins relative to the intact ACT. This allowed us to rule out that the inhibition in the lytic activity caused by the F521A and F532A mutations was due to a lower protein binding. Similarly, we could discard a greater binding as possible cause of the observed increment in the haemolysis percentage observed for the F415A, F485A mutant toxins. These binding data suggested that ACT association is mediated likely by multiple binding sites, and so that individually, none of the mutated CRAC/CARC sites avoided overall ACT binding to the membrane.

We explored further the effect on the ACT lytic capacity of double alanine substitutions in the two central phenylalanine residues of the two CRAC/CARC pairs (F415A-F485A and F521A-F532A mutants). The F521A-F532A double mutation reduced approximately to the half the maximum haemolysis percentage with respect to ACT (Figure 8). This reduction was similar to the observed for the single F521A or F532A mutations, indicating that the double mutation had no additive effect, and suggesting that it is enough the loss of one of both cholesterol-binding sites at the pore-forming domain to hinder formation of ACT pores. By contrast, the F415A-F485A double mutation had a clear additive effect, inducing a more potent increment of the maximum haemolysis, doubling the effect induced by either of the two single mutants, F415A or F485A (Figure 8).

Together, we had thus evidence that the four motifs explored are real, functional cholesterol-binding sites. Our results probed that cholesterol recognition through the CRAC^521^ and CARC^532^ motifs is involved and necessary for the pore-forming activity of ACT, very likely by driving the transmembrane insertion of two of the α-helices, HI and HII, of the pore-forming domain, that form part of the pore structure. The data suggest as well that interaction of h1 and h2, the two α-helices of the translocation region, with membrane cholesterol hinders the ACT lytic activity, since by preventing the interaction of the CARC^415^ and CRAC^485^ motifs with the sterol the haemolysis induced by the toxin is enhanced.

### 2.4. Substitutions by Ala of the Central Phe in 415, 485, 521 and 532 Positions, in the Respective Cholesterol-Recognition Motifs of ACT, Inhibit Prominently AC Domain Translocation

To determine whether the four CRAC/CARC motifs identified have any role in AC domain delivery, we measured in J774A.1 cells the effect on cAMP production of the Ala substitution in the central Phe residues 415, 485, 521 and 532 of the respective mutant proteins. As reflected in Figure 9, relative to intact ACT, the four single mutations F415A, F485A, F521A and F532A, prominently impacted the capacity by the respective mutant proteins to deliver the AC domain into the cytosol of the J774A.1 macrophages. Production of cAMP by all these mutants was indeed reduced drastically in a range of toxin concentrations between 25–200 ng/mL, with a greater inhibition observed for the F521A and F532A mutants. These results indicated that interaction with cholesterol through the four CRAC/CARC sites is necessary and instrumental to translocate the AC domain into the target cells and suggests that both the TR and the pore-forming domain must be inserted into the membrane for AC delivery.

## 3. Discussion

Insertion and assembly of PFT have been the subject of growing investigations to understand protein conversion from water-soluble forms to stable membrane-integrated structures [1,2,3,4], especially considering the resemblance of their mechanism of action to proteins of the vertebrate immune system, or to amyloid proteins [45,46]. Cholesterol, an essential component of the plasma membrane of eukaryotic cells, has been reported to have a crucial role in facilitating structural rearrangements of proteins upon association with the lipid bilayer [37]. In this study, we have made two major observations: first, we find that ACT specifically binds to membrane cholesterol; and second, we reveal the existence of four functional cholesterol-binding motifs in key membrane-interacting ACT domains, that may be molecular determinants of such cholesterol binding, thereby explaining the cholesterol-dependence of ACT cytotoxicity.

It was known that ACT lytic activity and AC delivery to target cell cytosol are modulated by the concentration of cholesterol in the cell membrane [30,31], but not whether this was due to direct binding to cholesterol, or to indirect effects of the sterol on the physical state of the phospholipid bilayer. Here we clarify this question by showing that ACT specifically recognizes cholesterol in membranes (Figure 1, Figure 2 and Figure 3). Further, we identify four cholesterol-binding motifs that are not conserved in the other toxins of the RTX family for which a specific interaction with cholesterol has been reported [33,34,35]. This may be related to the singularity of ACT of having an N-terminal domain with adenylate cyclase catalytic activity, besides the homologous C-terminal RTX haemolysin domain. Indeed, two of the identified motifs, the CARC^415^ and CRAC^485^ sites, are localized in two α-helices (h1 and h2) constituting the translocation region (see Figure 4), a segment adjacent to the AC domain that is not present in the rest of the RTX toxins and is necessary for AC delivery into target cells [28]. The other two motifs identified, the CRAC^521^ and CARC^532^ sites, localize in the first two helices (HI and HII) of the pore-forming domain, adjacent to the translocation region (see Figure 4), that insert into the lipid bilayer for building functional ACT pores [21,22,23,24,25,26,27]. Up to now, it was unknown whether insertion of these helices into the lipid bilayer is a thermodynamically spontaneous process or requires assistance from membrane lipid components. Neither was clear, which the exact topology of these helices is in the active ACT form.

Computer-assisted analysis of the ACT amino acid sequence for transmembrane helices using common prediction programs (not shown), as well as abundant mutational data by others [23,24,27] converge all on the prediction of an intramembrane location of the hydrophobic HII helix (residues 528–552) whereas predictions for HI helix are not unanimous. HI (residues 500–522) is amphipathic and presents two negatively charged Glu residues (E509 and E516) in the middle of its sequence, which could impose an energetic penalty, making transmembrane insertion of HI helix less favourable. The Philius algorithm (https://topcons.cbr.su.se/, accessed on 20 February 2021) predicts HI and HII to be both transmembrane, HI inserting with N_out_→C_in_ topology and HII with C_out_→N_in_ topology (not shown). According to this prediction the C-terminal CRAC^521^ of HI and the N-terminal CARC^532^ of HII would be placed in a membrane environment, both facing the cytosolic side of the plasma membrane, and five residues R^523^GSSR^527^ connecting both helices, would be outside the membrane (see Figure 10).

The ability of the CRAC/CARC sites to bind to cholesterol would be given by the structural characteristics of cholesterol, with a flat α-face and a rough β-face where the aliphatic groups are located (two methyl groups and an iso-octyl group), and by the characteristics of the CRAC/CARC sequences. In the linear CRAC/CARC sequences, the side chain of hydrophobic residues such as leucine and valine, can intercalate with the aliphatic chains of the rough face of cholesterol, being particularly well suited to contact with this β face of cholesterol through van der Waals forces [37]. The side chain of the central residue (Y/F) can interact with the α-face of cholesterol through CH-π interactions [47]; the positively charged Arg residues would locate their long non-polar aliphatic side chain within the apolar membrane, and the basic positively charged group emerging at the interface of the membrane [48]. Our data showing that single substitutions of the Phe residues 521 and 532 by Ala in the CRAC^521^ and CARC^532^ motifs cause a potent inhibition of the ACT activity (Figure 6, Figure 8 and Figure 9) are fully consistent with the notion that these two central aromatic residues are buried within the membrane and participate in cholesterol binding. These data thus provide experimental support to the aforementioned Phillius prediction that places both HI and HII in transmembrane location and suggest that binding to membrane cholesterol through the CRAC^521^ and CARC^532^ motifs is involved in the stabilization of HI and HII within the lipid bilayer and conferring the adequate membrane topology for both haemolysis and AC translocation.

Recently the ACT segment linking the AC domain to the RTX haemolysin moiety (residues 400–500) and constituted by two α-helices (h1 and h2) was reported to be necessary for AC translocation, since deletion of residues 375–485 within ACT totally abrogated the toxin’s ability to increase intracellular cAMP in target cells. However, so far, it was not clear whether both h1 and h2 interact with the membrane for translocation or only h2 [49]. Using two synthetic peptides corresponding to each one of the helices, one group recently noted that only the peptide mimicking h2 binds to membranes containing anionic lipids, adopting an α-helical structure oriented in plane with respect to the lipid bilayer, whereas the peptide mimicking h1, does not interact [49]. Our results with the two mutants in the central Phe residue of the CARC^415^ and CRAC^485^ motifs (F415A and F485A mutants) identified in the translocation region have been very revealing at this regard (Figure 6, Figure 8 and Figure 9). They show for one side that both substitutions (individually or combined in a double mutant) cause a prominent inhibition of the ACT capacity to generate cAMP in target cell cytosol, and for other side, augment simultaneously the toxin lytic capacity. This suggests that in cholesterol-rich membranes such as the eukaryotic plasma membrane, binding to cholesterol through the CARC^415^ and CRAC^485^ motifs drives h1 and h2 to embed into the membrane, and further, that this insertion is essential for AC translocation. This is consistent with a previous study showing that a monoclonal antibody 3D1, which binds to an epitope (amino acids 373 to 399) at the distal end of the catalytic domain of ACT, blocks AC delivery and causes an increase in the haemolytic activity of three to four folds [50]. By contrast, our data seems to disagree with the view that only h2 interacts with the membrane [49]. The apparent discrepancy may be due to that none of the two peptides used by the aforementioned group contained in their sequence the corresponding cholesterol-recognition motif CARC^415^ and CRAC^485^.

Having established the membrane topology of the HI and HII helices of the pore-forming domain (Figure 10) and basing on the data with the CARC^415^ and CRAC^485^ mutants that support an intramembrane insertion of h1 and h2 helices, the topology of these last two helices can be delineated. h2 would insert with C_out_→N_in_ orientation and h1 with N_out_→C_in_ orientation. In this case, thus the CARC^415^ and CRAC^485^ motifs would be both facing the extracellular side, binding to cholesterol molecules in the exofacial side of the membrane. The transmembrane topology of the h1-h2-HI-HII α-helices, as proposed here is schematically depicted in Figure 11. Given that functional ACT lytic pores, besides the aforementioned HI and HII helices, involve the insertion into the membrane of the helices HIII to HV of the pore-forming domain [21,22,27,44], a membrane topology can be reasonably proposed for these last ones by extension of HI-HII topology, getting this way a more complete picture of the putative topology of the set of helical elements involved in AC translocation and in building the ACT pore structure (Figure 12).

Interestingly, from this resultant membrane topology it emerges that ACT molecule would adopt one single conformation to accomplish both AC transport and lytic activity, challenging the previously assumed model of conformational duality of ACT [21,22,27]. That model postulated that two toxin conformers in equilibrium would be involved in ACT activities, one ACT conformer leading to the direct AC transport across the lipid bilayer, and other one, leading to pore formation [21,22,27].

For long, ACT pores have been regarded as too small (0.6–0.8 nm in diameter) for the passage of even an unfolded polypeptide chain [19]. This directly led to discard the possibility that the pore formed by this toxin might serve to transport the AC domain to the target cytosol [19]. Contrasting with the small pore view, our laboratory has recently revealed that the ACT pores are of proteolipidic nature, involving lipid molecules besides segments of the protein lining the pore walls [20]. As consequence of this more dynamic structure of the ACT pore, it is anticipated that its hydrophilic lumen may be wider than previously believed. From the conjunction of these two points, single transmembrane topology (Figure 12) and wider ACT pore [20], a plausible mechanism for AC translocation emerges by which the 400-residue-long AC polypeptide would be transported to the cytosol of the target cell directly through the hydrophilic “hole” formed by the ACT pore. We hypothesize that cholesterol-mediated anchoring to the membrane of the h1-h2 helices would allow the N-terminal AC domain to be close to the pore structure (helices I–V). This spatial proximity from the pore could allow interactions to be stablished between residues of the AC domain and residues at the pore entrance, guiding the AC polypeptide to penetrate into the pore. Membrane potential, reported to be necessary for translocation [51], might then provide energy required for pulling down the polypeptide along the pore lumen. Then as the AC polypeptide leaves the pore and exits to the cytosol by its N-terminus it might be recognized by calmodulin. Reasonably, any mutation that affects cholesterol binding, and hence insertion of the helices, will have a deleterious effect on AC translocation, in full consonance with our present results. On other side, preventing the insertion of the h1-h2 helices into the membrane would also avoid the approaching of the AC domain to the pore structure, which could make the steric hindrance at the pore entrance to disappear, allowing a free ion flux, which would be experimentally detected as a greater haemolysis. This is fully consistent with the observed here for the F415A and F485A mutants, an increase in lytic activity in parallel to the inhibition of the AC domain translocation. The “pore model” for AC translocation proposed here may thus explain why ACT is apparently weakly haemolytic relative to other RTX pore-forming toxins, such as *Escherichia coli* α-haemolysin, which do not possess an equivalent N-terminal domain [18,19]. Supporting this idea that in native conditions the presence of the AC domain somehow blocks the ion flux through the ACT pore, others previously observed that the elimination of the AC domain plus residues of the translocation region from the ACT polypeptide made the mutant toxins to exhibit a lytic potency comparable to the RTX toxins [52,53]. Future work will be needed to further prove this effect and map interactions between the AC domain and residues at the pore entrance.

We believe that our study adds new relevant insights to a field with scarce structural and mechanistic information and may thus inspire other investigators to raise new questions that can pave the way for understanding the mechanisms of cytotoxicity by ACT and other RTX toxins, and hopefully to have a 3D structure of ACT on lipid bilayers.

In this study, we unveil that direct ACT interaction with membrane cholesterol through four cholesterol-binding motifs we identify in ACT sequence is instrumental for stabilizing the proper transmembrane topology of a set of helices forming part of the translocation and pore-forming domains necessary for ACT activity. To our best knowledge, the here presented model of membrane topology accounts for all available experimental data and suggests a plausible mechanism by which ACT can translocate the AC domain on target cells, at the cost of sacrificing the lytic potency. Given the relevance of the specific cholesterol-recognition sites in ACT activity, it can be anticipated that targeting the here identified four CRAC/CARC motifs could be a new therapeutic option for inhibiting cholesterol binding and hence reducing the toxicity of ACT on cells.

## 4. Materials and Methods

### 4.1. Expression and Purification of Intact ACT

ACT was expressed in *Escherichia coli* XL-1 blue cells (Stratagene) transformed with pT7CACT1 plasmid, kindly provided by Dr. Peter Sebo (Institute of Microbiology of the ASCR, v.v.i., Prague, Czech Republic) and purified as described by Karst et al. [54].

### 4.2. Construction, Expression and Purification of the ACT Mutants F415A, F485A, F521A and F532A

The variants of ACT F415A, F485A, F521A and F532A were cloned, expressed and purified from E. coli. cyaA DNA was amplified from genomic DNA by PCR and cloned in pET-15b (GenScript Biotech, 2288 EG Rijswijk, The Netherlands) using AsuII and NcoI enzymes to generate plasmid pME14. Site-directed mutagenesis according to Agilent protocol was performed on pME14 to replace Ala codons for Phe in 415, 485, 521 and 532 residues. All plasmid inserts were sequenced to confirm accuracy of PCR and mutagenesis. For protein expression, *E coli.* BL21 transformed with pME14 plasmid was grown in LB with 100 μg mL^−^^1^ ampicilin to A_600_ = 0.6–0.8 and protein expression was induced by 4 h growth in 1 mM isopropyl-β-D-thiogalactopyranoside. Protein purification was performed according to the method described in Karst et al., (2014) [54]. Concentrations of purified ACT proteins were determined by the Bradford assay (Bio-Rad, Hercules, CA, USA) using bovine serum albumin as standard. All toxins purified by this method were more than 90% pure as judged by SDS-PAGE analysis (not shown).

### 4.3. Haemolysis Assay

Haemolysis assays were performed on 96-well plates. Briefly, serial dilutions of ACT (starting at 50 nM) in assay buffer (20 mM Tris pH 8.0, 150 mM NaCl, 2.0 mM CaCl) were prepared, onto which an equal volume of erythrocytes at a density of 5 × 10^8^ cells/mL were added, and the mixtures incubated at 37 °C for 180 min under constant stirring. At the end of the incubation time, the plates were centrifuged (2000× *g*, 10 min, 4 °C) and the supernatant scattering was measured at 700 nm. Alternatively, time course experiments were performed recording continuously the scattering signal at 700 nm. The blank (0% haemolysis) corresponded to erythrocytes incubated in buffer without toxin and 100%, and 100% haemolysis was obtained by adding Triton X-100 (0.1%) to the erythrocyte suspension.

### 4.4. Cell Culture

J774A.1 macrophages (ATTC, number TIB-67) were grown at 37 °C in DMEM (Sigma Aldrich, St. Louis, MO, USA) containing 10% (*v/v*) heat inactivated FBS (Thermo Fisher Scientific, USA), 6 mM L-glutamine (Thermo Fisher Scientific, Waltham, MA, USA), 0.2% (*v/v*) MycoZap Prophylactic (Lonza, Switzerland) and Penicillin-Streptomycin (Sigma Aldrich, USA) (100 U/mL and 100 µg/mL respectively) in a 90% humidified atmosphere with 5% CO_2_.

### 4.5. Measurement of cAMP

cAMP produced in cells was measured upon incubation of different ACT concentrations (25–200 ng/mL) with J774A.1 cells (5 × 10^5^ cells/mL) for 30 min at 37 °C. cAMP production was calculated by the direct cAMP ELISA kit (Enzo Lifesciences, Farmingdale, NY, USA). Absorbance at 405 nm was measured for determinations.

### 4.6. Measurement of ACT or Mutant Toxins Binding to Lipid Membranes Determined by Flotation Assays

Membrane association of ACT or ACT variants was assayed by flotation assay using large unilamellar vesicles (LUVs). LUVs were prepared following the extrusion method of Hope et al. [55]. Phospholipids and cholesterol were mixed in chloroform and dried under a N_2_ stream. Traces of organic solvent were removed by 2 h vacuum pumping. Subsequently, the dried lipid films were dispersed in buffer and subjected to 10 freeze-thaw cycles prior to extrusion 10 times through 2 stacked polycarbonate membranes with a nominal pore size of 100 nm (Nuclepore, Inc., Pleasanton, CA, USA). Phospholipid concentration of liposome suspensions was determined by phosphate analysis [56]. Liposome size was determined by Dynamic Light Scattering in Zetasizer Nano ZS (Malvern Panalytical Ltd., Malvern, UK). Vesicle flotation experiments in sucrose gradients were subsequently performed following the method described by Yethon et al. [57]. In brief, 750 nM ACT and 1.5 mM LUVs (DOPC and DOPC.Chol 3:1 molar ratio, with 0.5% Rhodamine) are incubated for 30 min at 37 °C, under stirring. 125 µL of each sample was adjusted to a sucrose concentration of 1.4 M in a final volume of 300 µL and subsequently overlaid with 400 µL and 300 µL layers of 0.8 and 0.5 M sucrose, respectively. The gradient was centrifuged at 436,000× *g* for 180 min in a TLA 120.2 rotor (Beckman Coulter, Brea, CA, USA). After centrifugation, four 250 µL fractions were collected. The material adhered to the tubes was collected into a fifth fraction by washing with 250 µL of hot (100 °C) 1% (*w/v*) SDS. The different fractions were run on SDS-PAGE, and the presence of ACT was probed by Coomassie. Liposomes were monitored by measuring rhodamine fluorescence. The values displayed on the right correspond to the percentages of protein found co-floating with vesicles, calculated by densitometry. Densitometry of the bands was performed by using ImageJ software, and the percentage of binding to vesicles was calculated from the band intensities measured in the vesicle-floating fractions, relative to the sum of the intensities measured in all fractions. The results displayed are representative of at least two replicates.

## Figures and Tables

**Figure 1 ijms-23-08703-f001:**
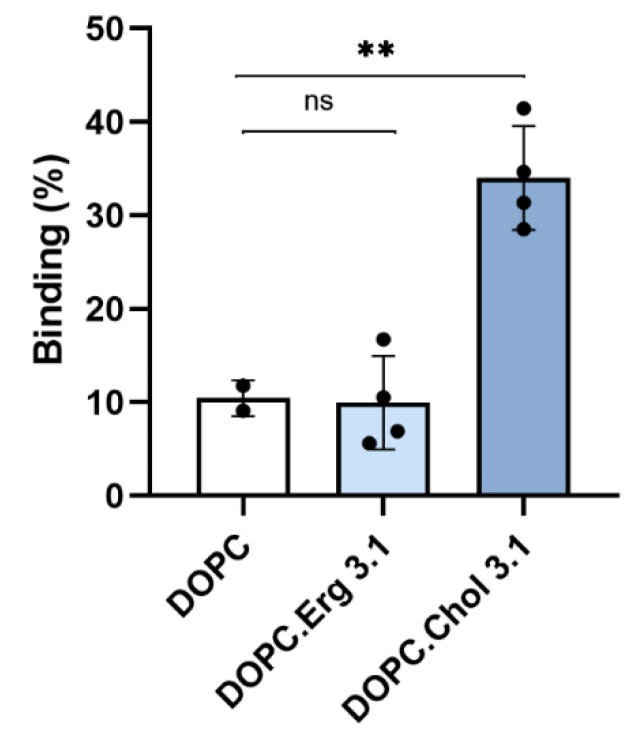
Binding of full length ACT to vesicles of different lipid composition. Membrane association of ACT to lipid bilayers of different lipid composition as measured by flotation assays using large unilamellar vesicles composed of DOPC, DOPC:Chol (1:1 molar ratio) or DOPC:Ergosterol (1:1 molar ratio). Details on the flotation assay methodology are provided in the Experimental Procedures section. The mean and standard deviations of three independent experiments are shown. Statistical differences are based on One-way ANOVA test with Dunnett’s T3 multiple comparisons; [ns] non-significant *p* ≥ 0.05 and ** *p* ≤ 0.01.

**Figure 2 ijms-23-08703-f002:**
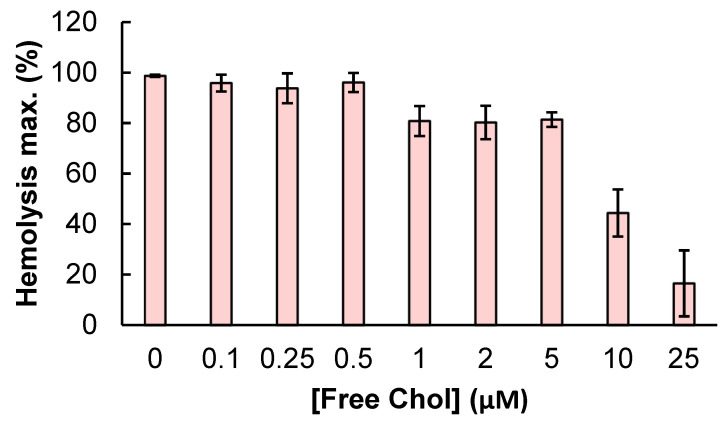
Effect of ACT preincubation with free cholesterol on ACT-induced haemolytic activity. ACT (100 nM) was preincubated for 30 min at RT in the presence of free cholesterol (0–25 µM). Then sheep erythrocytes at a density of 5 × 10^8^ cells/mL were added and the mixture was further incubated for 180 min at 37 °C. Haemolytic activity was measured as decrease of turbidity at 700 nm and expressed as haemolytic percentages (calculated as detailed in the Experimental Procedures section). Data represented in the figure correspond to the mean of three independent experiments ± SE.

**Figure 3 ijms-23-08703-f003:**
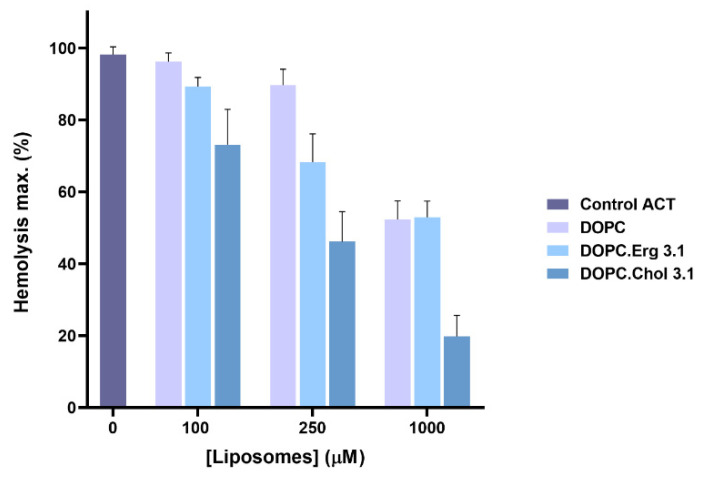
Effect of ACT preincubation with liposomes of different lipid composition on ACT-induced haemolytic activity. ACT (50 nM) was preincubated for 30 min at RT in the presence of liposomes of different lipid compositions (DOPC, DOPC:Chol 3:1 and DOPC:Erg 3:1 molar ratio) at different total lipid concentrations (0–1000 µM). Then sheep erythrocytes at a density of 5 × 10^8^ cells/mL were added and the mixture was further incubated for 180 min at 37 °C. Haemolytic activity was measured as decrease of turbidity at 700 nm and expressed as haemolytic percentages (calculated as detailed in the Experimental Procedures section). Data represented in the figure correspond to the mean of three independent experiments ± SE.

**Figure 4 ijms-23-08703-f004:**
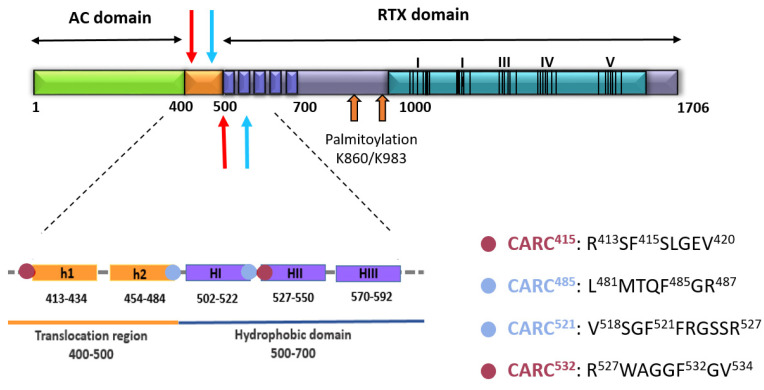
Schematic drawing of the ACT polypeptide chain with the identified four potential cholesterol-recognition motifs. ACT is a 1706-residue-long polypeptide that consists of an N-terminal adenylate cyclase enzyme domain (AC domain, residues 1 to ≈400) (in green) that is fused by an “AC to Hly linker segment” of about 100 residues (in orange) so-called translocation region (residues ≈400–500) to a pore-forming RTX haemolysin (Hly) moiety of approximately 1200 residues (different blue tones). The RTX haemolysin moiety in turn, contains a hydrophobic pore-forming domain comprising residues 500 to 700, constituted by five alpha-helices (dark blue-coloured five cylinders), an acylated domain between residues 800 and 1000, where the posttranslational acylation at two lysine residues (K860 and K983, two orange arrows) occurs, a typical calcium-binding repeats domain (in light blue) organized in five blocks (I to V) and a C-terminal secretion signal (last ≈100 residues). Two predicted α-helices in the translocation region, namely, h1 and h2 (in orange), and three of the five predicted amphipathic/hydrophobic helices of the pore-forming domain, namely HI, HII and HIII (in dark blue) have been depicted below with greater detail. Blue or red spots have been used in the scheme below to specify the location of each one of the four potential cholesterol-recognition motifs identified in this study (CARC^415^, CRAC^485^, CRAC^521^ and CARC^532^). Sequences of each one of the motifs are specified on the right side of the scheme. The respective N-terminal leucine/valine or arginine, the central phenylalanine (F415, F485, F521 and F532) and the C-terminal arginine or valine residues of the predicted CRAC and CARC motifs are specified. The four motifs are also indicated with blue and red arrows in the schematic drawing of the ACT structure.

**Figure 5 ijms-23-08703-f005:**
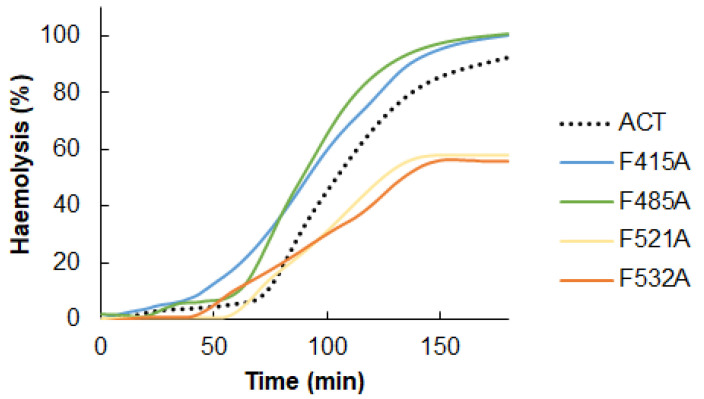
Effect of Ala substitutions in the central Phe residues of the potential cholesterol-binding sites CRAC^415^, CARC^485^, CRAC^521^ and CARC^532^ on the kinetics of the ACT-induced haemolysis. Raw traces of the kinetics recorded from a representative experiment of the haemolysis induced by intact ACT (50 nM) or by each one of the four mutant toxins (50 nM). A suspension of sheep erythrocytes (5 × 10^8^ cells/mL) was incubated with each protein for 180 min at 37 °C, recording the scattering changes measured at 700 nm at every second. Then the haemolysis percentage was calculated as detailed in the Experimental Procedures section and depicted in the figure. The traces shown correspond to a representative experiment from three experiments performed independently.

**Figure 6 ijms-23-08703-f006:**
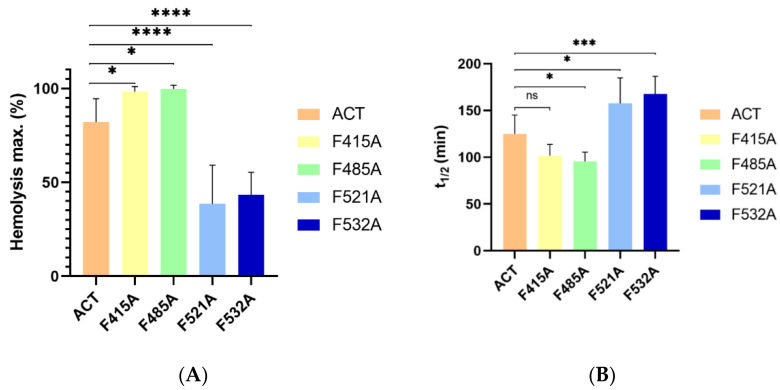
Effect of Ala substitutions in the central Phe residues of the potential cholesterol-binding sites CRAC^415^, CARC^485^, CRAC^521^ and CARC^532^ on the (**A**) maximum haemolytic percentage, and (**B**) t_1/2_ of the ACT-induced haemolysis. Haemolysis induced by 50 nM of intact ACT or by each one of the four mutant toxins was assayed with a suspension of sheep erythrocytes (5 × 10^8^ cells/mL) incubated with each protein for 180 min at 37 °C. Data represented in the figure correspond to the mean of three independent experiments ± SE. *p*-values for the plot in the left subpanel * *p*  = 0.0444 (ACT/F415A); * *p*  = 0.0262 (ACT/F485A); **** *p* < 0.0001; *p*-values for the plot in the right subpanel * *p*  = 0.0202 (ACT/F485A); * *p*  = 0.047 (ACT/F521A); *** *p* = 0.0008.

**Figure 7 ijms-23-08703-f007:**
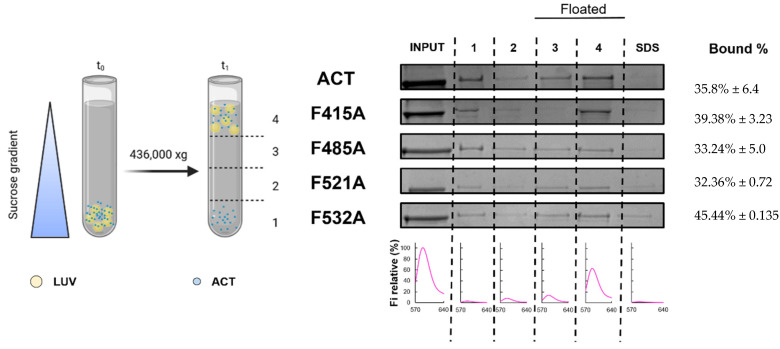
Quantification of the binding of ACT or ACT mutants to lipid bilayers. Membrane partitioning as measured by flotation assays using large unilamellar vesicles composed of DOPC:Chol (3:1 molar ratio). Details on the flotation assay methodology are provided in the section of Experimental Procedures. Bound % data correspond to the mean of three independent experiments ± SE.

**Figure 8 ijms-23-08703-f008:**
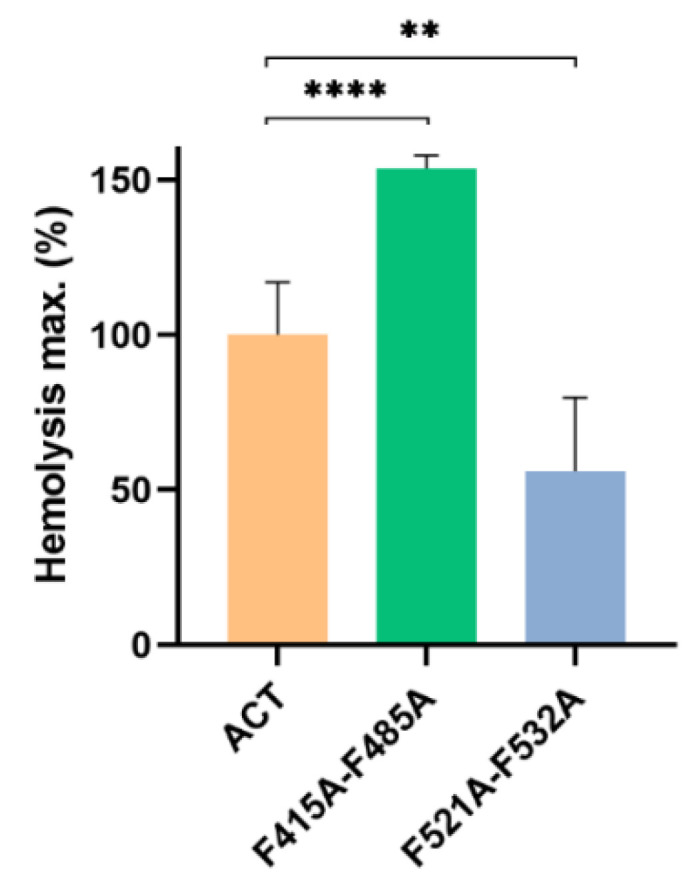
Effect of double Ala substitutions in the central Phe residues of the potential cholesterol-binding sites F415A-F485A and F521A-F532A on the maximum haemolytic percentage. Haemolysis induced by 50 nM of intact ACT or by each one of the mutant toxins was assayed with a suspension of sheep erythrocytes (5 × 10^8^ cells/mL) incubated with each protein for 180 min at 37 °C. Data represented in the figure correspond to the mean of three independent experiments ± SE. The one-way ANOVA (Brown–Forsythe test) with Dunnett multiple comparisons test was used to determine whether there is a significant difference between the mean values of our independent groups (** if *p* ≤ 0.01 and **** if *p* ≤ 0.0001).

**Figure 9 ijms-23-08703-f009:**
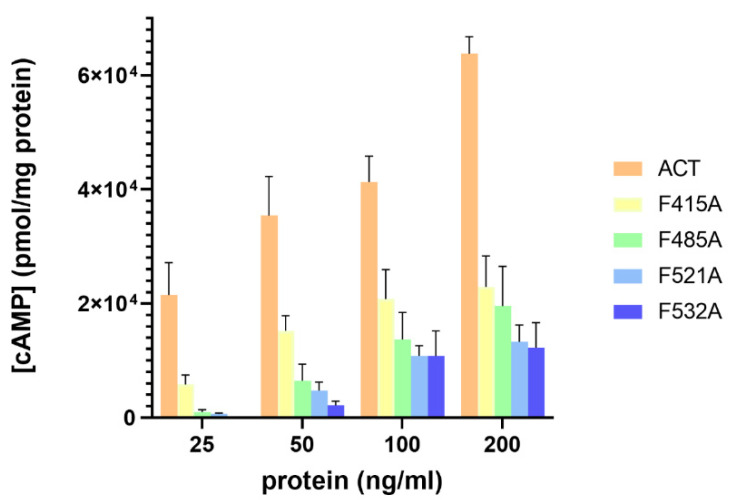
Point Ala substitutions in the central Phe residue of the potential cholesterol-binding sites CRAC^415^, CARC^485^, CRAC^521^ and CARC^532^ prominently decrease AC domain translocation. Translocation of AC domain was assessed by determining the intracellular concentration of cAMP (pmol/mg protein), generated in J774A.1 cells (1 × 10^5^ cells/mL) suspended in 20 mM Tris-HCl, pH = 8.0 buffer, supplemented with 150 mM NaCl and 2 mM CaCl_2_. Cells were treated for 30 min at 37 °C with different concentrations (25–200 ng/mL) of intact ACT, or the corresponding mutant toxin. Data represent the mean ± SD of at least three independent experiments.

**Figure 10 ijms-23-08703-f010:**
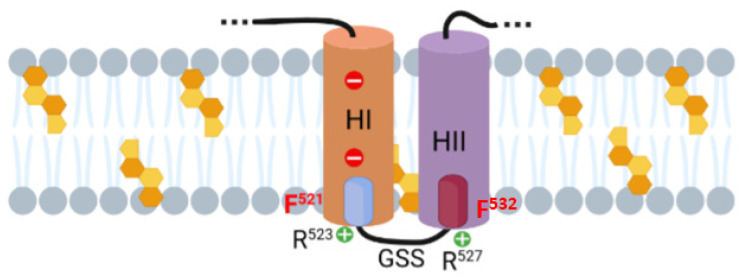
Schematic representation of the proposed membrane topology for HI and HII helices of the pore-forming domain of ACT. Proposed topology for the HI and HII helices as predicted by the algorithm Philius (https://topcons.cbr.su.se/, accessed on 20 February 2021); the two cholesterol recognition motifs identified in this study, CARC^532^ and CRAC^521^ motifs, are represented by blue and purple cylinders, respectively, and the central aromatic residue of each site (F521 and F532) is highlighted in red. Cholesterol molecules are represented by orange-coloured penta-hexagonal figures. More details are explained in the Discussion section. Original picture created with BioRender.com.

**Figure 11 ijms-23-08703-f011:**
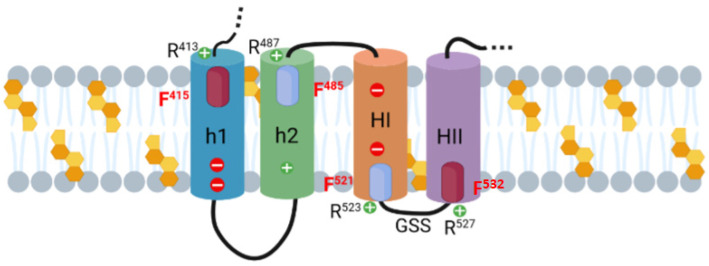
Schematic model of the membrane topology for the h1-h2-HI-HII helices. The figure shows a scheme of the membrane topology for the h1, h2 (TR region) and HI-HII (pore-forming domain) helices as proposed here. Assuming that HI and HII would adopt N_out_→C_in_ and C_out_→N_in_ topology, respectively (as explained in the text), then, h2 and h1 topology would be C_out_→N_in_ and N_out_→C_in_, respectively, with the difference that in this case the CARC^415^ and CRAC^485^ motifs would be facing the extracellular side, and so h1 and h2 would bind cholesterol in the exofacial, side of the membrane. The four cholesterol-binding sites identified in this study, the CRAC^415^ and CRAC^532^ motifs (purple cylinders) and the CARC^485^ and CARC^521^ motifs (blue cylinders) and their respective central phenylalanine residues F415, F485, F521 and F532 (in red) are highlighted. Other residues cited in the text, in the Discussion section, have been included. Cholesterol molecules are represented by orange-coloured penta-hexagonal figures. Original picture created with BioRender.com.

**Figure 12 ijms-23-08703-f012:**
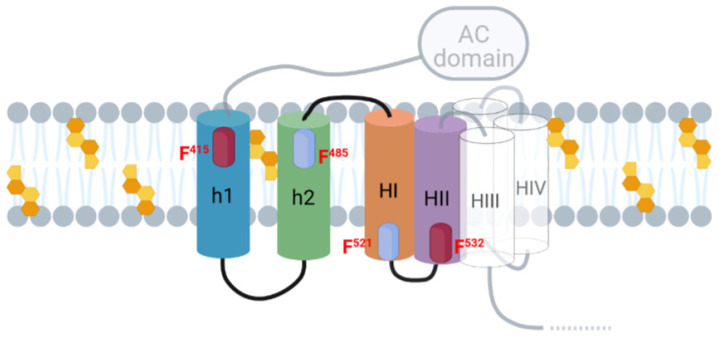
Schematic model of the membrane topology for the translocation region and the pore-forming domain. The figure shows a scheme of the putative membrane topology for the h1, h2 (TR region) and HI-HV (pore-forming domain) helices involved in AC translocation and in building the ACT pore structure as proposed here. Original picture created with BioRender.com.

**Table 1 ijms-23-08703-t001:** Cholesterol-recognition motifs identified in the sequence of the membrane-interacting translocation region and hydrophobic domain of ACT.

Pattern		Amino Acids	Sequence
[**L/V]-X(1,5)-Y-X(1,5)-[R/K]**	**HYDROPHOBIC DOMAIN**	626–638	**L**VQQSH**Y**ADQLD**K**
		653–661	**L**LAQL**Y**RD**K**
		721–728	**L**AND**Y**AR**K**
		732–741	**L**GGPQA**Y**FE**K**
**[L/V]-X(1,5)-F-X(1,5)-[R/K]**	**TRANSLOCATION REGION**	481–487	**L**MTQ**F**G**R**
	**HYDROPHOBIC DOMAIN**	518–527	**V**SG**F**FRGSS**R**
**[R/K]-X(1,5)-F-X(1,5)-[L/V]**	**TRANSLOCATION** **REGION**	413–420	**R**S**F**SLGE**V**
	**HYDROPHOBIC DOMAIN**	527–534	**R**WAGG**F**G**V**

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
