# Peer review of "Four Cholesterol-Recognition Motifs in the Pore-Forming and Translocation Domains of Adenylate Cyclase Toxin Are Essential for Invasion of Eukaryotic Cells and Lysis of Erythrocytes"

_ijms, 2022, doi:10.3390/ijms23158703_

Round 1

Reviewer 1 Report

Comments to Authors

            Amuategi et al. reported in this study that ACT specifically interacts with membrane cholesterol, and four cholesterol-binding motifs in two membrane-interacting ACT domains are essential for AC domain translocation and lytic activities. They also speculated how ACT exerts its toxicity by modeling the topology of several transmembrane regions of ACT. The manuscript is fundamentally interesting and could potentially provide a clue to understanding the mechanisms of cytotoxicity by ACT and other RTX toxins. However, some points in this manuscript should be revised. 

Major point

            Discussion section is redundant. The authors propose mechanisms of cholesterol-dependent cytotoxicity in the discussion, many of which do not appear to be directly demonstrated or inferred from the present data. For example, the AC domain migrates into the cytoplasm through a pore formed by the helix HI-V (lines 469-487), and the E509K and E516K mutants in the HI helix invert the membrane topology of HI, which in turn reverses HII-V (lines 518-538) are not beyond the realm of imagination. It seems not beyond the realm of imagination. If the former, they should show an interaction between the AC domain and the HI-V complex.

Minor points

1)    Figure 1: Why do the authors use not One-way ANOVA and post-hoc test but a two-tailed homoscedastic t-test ?

2)    p.4 Line 152. Omit an underline of ° (37°C).

3)    Figure 4: What is the meaning of the diagonal dotted line? If you don't need it, remove it.

4)    p.10 Line 357-360. "second, we reveal the existence of four functional cholesterol-binding motifs in key membrane-interacting ACT domains that may me molecular determinants of such cholesterol-binding, thereby explaining the cholesterol-dependence of ACT cytotoxicity." Is the verb which is in red correct?

Author Response

Please find the attachment below with the point-by-point response to the reviewer´s comments.

Reviewer 2 Report

The article “Four cholesterol-recognition motifs in the pore-forming and translocation domains of Adenylate Cyclase Toxin are essential for invasion of eukaryotic cells and lysis of erythrocytes” analyzes the mode of interaction of Adenylate Cyclase Toxin from Bordetella pertussis with the cell membrane.

The authors have shown a great experimental work. The introduction collects previous studies and is displayed appropriately

Although the authors clearly present their results, the article can be improved on some points.

-        Regarding the presentation of the results in Table 1, a table header should be written. With respect to Fig 3, it would be convenient to name the axis “x” as in fig. 9 and 2; and thus a homogeneous structure would be maintained in the article.

-        In fig.4 you cannot see what is reflected in the figure caption

-        Fig 10 is not clear imagen

Regarding the methodology section,

-        I see the need to explain in more detail the subsection “Construction, expression and purification of the ACT mutants F415A, F485A, F521A and 580 F532A”; Since the article talks about 4 mutants, the primers used both in PCR and in Site-directed mutagenesis should be reflected. The authors refer to the Agilent protocol, but since the article is based on these mutants, it could be briefly reflected.

-        "Measurement of cAMP", the absorbance used to determine the amount of cAMP was not reflected

At the references section, the same style is not followed when citing different previous works

And finally, a section of Conclusions should be reflected where some more detail is reflected that should be highlighted from the discussion.

Author Response

Please, find the attachments below with the point-by-point response to the reviewer´s comments.

Reviewer 3 Report

This manuscript presents four cholesterol-recognition motifs in the adenylate cyclase toxin pore-forming domain and translocation domain that are important for the invasion of eukaryotic cells. The authors use site-directed mutagenesis to study the roles of four cholesterol recognition amino acid consensus motifs (CRAC or CARC) in pore-forming and translocation domains. It is known that cholesterol is critical for ACT’s biological activity. The aromatic amino acid in CRAC is required for cholesterol binding. It is interesting that despite the increase in haemolysis by F415A and F485A, they decrease the AC domain translocation. Authors discussed extensively the different proposed models, however, those models (Figures 10-13) are just hypothetical without any structural confirmations and seem overreached. Therefore, it’d be better to discuss just based on data they have unless they design more experiments to prove those models.

Other comments: Authors should use more clear language in the manuscript. Some sentences are confusing and difficult to understand. Some are typos (for example, in line 359, “may me” should be “may be”).

Overall, the authors need to do some major revisions on their discussion part, especially reducing the lengthy hypothetical discussion, and focusing more on their data.

Author Response

(The authors gave the same response as above.)

Round 2

Reviewer 1 Report

Thank you for your email, but my email adress was updated at this April.
Thus, your email was not delivered or delayed for recognition as spam.

I  have already reviewed the revised manuscript.
However, I am now busy and I could not reply.

I think that the discussion section is still long, but improved.
I agree for publication in this manuscript. 

Reviewer 3 Report

It can be accepted after the editorial review.